# Reduced Activity of Soluble Fibroblast Activation Protein (sFAP) Represents a Biomarker of Aggressive Disease in Lymphoid Malignancies

**DOI:** 10.3390/ijms262311248

**Published:** 2025-11-21

**Authors:** Jonas Klejs Hemmingsen, Marie Hairing Enemark, Anne Kathrine Nissen Pedersen, Emma Frasez Sørensen, Kristina Lystlund Lauridsen, Julie Bondgaard Løhde, Francesco d’Amore, Stephen Jacques Hamilton-Dutoit, Mette Bjerre, Maja Ludvigsen

**Affiliations:** 1Department of Hematology, Aarhus University Hospital, 8210 Aarhus, Denmark; 2Department of Clinical Medicine, Aarhus University, 8210 Aarhus, Denmark; 3Medical/Steno Aarhus Research Laboratory, Department of Clinical Medicine, Aarhus University, 8210 Aarhus, Denmark; 4Department of Pathology, Aarhus University Hospital, 8210 Aarhus, Denmarkshdutoit@gmail.com (S.J.H.-D.)

**Keywords:** fibroblast activation protein (FAP), lymphoid malignancies, soluble biomarker, tissue expression, time-resolved immunofluorometric assay (FRET), time-resolved immunofluorometric assay (TRIFMA)

## Abstract

Fibroblast activation protein (FAP), a transmembrane serine protease expressed primarily in pathological conditions, plays a pivotal role in tumor progression. Despite extensive studies on FAP in solid tumors, its role in hematologic cancers, particularly lymphoid malignancies, remains underexplored. This study aimed to investigate the level and activity of soluble FAP (sFAP) in pre-therapeutic serum samples from 120 lymphoma patients. We measured sFAP serum levels using time-resolved immunofluorometric assay and sFAP activity with Förster resonance energy transfer assay. In addition, immunohistochemistry was used to analyze intratumoral FAP expression in tissue biopsies from a subset of B-cell lymphoma patients (*n* = 34). Notably, the results revealed significantly reduced circulating sFAP levels (*p* = 0.002) and activity (*p* < 0.001) in aggressive disease subtypes compared with indolent subtypes and healthy individuals. At the time of diagnosis, low sFAP activity correlated with inferior overall survival (both *p* < 0.001) in patients with the aggressive entities, suggesting altered FAP functionality in these tumors. Interestingly, measuring intratumoral FAP levels revealed an inverse pattern, with diffuse large B-cell lymphoma showing higher tissue FAP localization compared with follicular lymphoma (*p* < 0.001). These findings provide new insights into the biological and clinical significance of FAP in lymphoid malignancies, particularly highlighting the importance of sFAP activity as a potential prognostic marker in aggressive lymphoid malignancies.

## 1. Introduction

Fibroblast activation protein (FAP) is transmembrane serine protease that has attracted increasing attention due to its expression being almost exclusive to pathological conditions, including fibrosis, arthritis, and cancer [1,2,3]. FAP possesses both dipeptidyl peptidase and endopeptidase activities, which enable it to cleave various substrates. Although its complete substrate profile remains largely unknown, several important targets have been identified, including neuropeptide Y, peptide YY, fibroblast growth factor 21 (FGF21), a2 anti-plasmin, and multiple collagen types [2,3].

In cancer, FAP has been implicated in a variety of tumor-promoting processes. Elevated FAP expression is associated with poor prognosis across several cancer types, including carcinomas, sarcomas, and other solid tumors [2,3,4]. This is particularly true in the tumor stroma, where FAP is commonly expressed on cancer-associated fibroblasts (CAFs), a major component of the tumor microenvironment; however, it is notably absent in normal fibroblasts [5]. Accordingly, there has been growing interest in exploiting FAP biology as a potential attractive target for clinical research. FAP is believed to contribute to tumor progression through its effects on extracellular matrix (ECM) remodeling, cell migration, invasion, and metastasis [6]. These processes are further facilitated by FAP’s ability to activate signaling pathways, such as PI3K and matrix metalloproteinases (MMP2/9), through direct protein interactions [2,7,8,9]. While FAP exhibits catalytic activity through its serine protease function, emerging evidence suggests that its influence on cell behavior also extends beyond enzymatic processes, potentially involving non-catalytic mechanisms [2]. Moreover, crosstalk between CAFs, cancer cells, and other stromal cells is vital for the acquisition of several hallmarks of cancer [2,5,10]. Importantly, FAP also plays a significant role in immune modulation within the tumor microenvironment. Studies have shown that FAP-expressing CAFs can promote immunosuppression by interfering with dendritic cell maturation, inhibiting T-cell activation, and suppressing the expression of major histocompatibility complex (MHC) molecules, thus impairing the body’s ability to mount an effective immune response [2,3,11].

While FAP expression in the stroma has been established in several cancer types, its soluble form, sFAP, also holds considerable interest. sFAP (also referred to as α2-antiplasmin cleaving enzyme (APCE)) is shed from the cell membrane and retains both dipeptidyl peptidase and endopeptidase activity, with a similar substrate profile to membrane-bound FAP [12,13,14].

Lymphoid malignancies, arising from immune cells, represent a unique group of cancers due to their diverse origins, molecular abnormalities, and clinical outcomes [15]. While FAP is somewhat well-studied in solid tumors, less is known about its role in lymphoid malignancies. In multiple myeloma, elevated FAP levels are correlated with poor outcomes. Changes in FAP expression have been linked to the β-catenin signaling pathway and thus potentially protects tumor cells from chemotherapy-induced apoptosis [2,16,17]. Similar findings in acute myeloid leukemia further underscore the importance of FAP as a potential indicator of adverse prognosis [18].

Despite increasing interest in FAP and sFAP, their roles in lymphoid malignancies remain poorly understood. To address this gap, the present study aimed to investigate the level and activity of sFAP in pre-therapeutic serum samples from a cohort of patients diagnosed with lymphoid malignancies. We utilized time-resolved immunofluorometric (TRIFMA) and Förster resonance energy transfer (FRET) assays to measure sFAP serum level and activity while characterizing pre-therapeutic tissue expression of FAP using immunohistochemistry (IHC) in a subset of the cohort.

## 2. Results

### 2.1. Patient Characteristics

This study included a total of 120 patients, diagnosed with a variety of different B- and T-cell malignancies, including diffuse large B-cell lymphoma (DLBCL, *n* = 30), follicular lymphoma (FL, *n* = 13), chronic lymphocytic leukemia/small lymphocytic lymphoma (CLL/SLL, *n* = 38), Hodgkin’s lymphoma (HL, *n* = 11), other B-cell nHLs (*n* = 21), and T-cell lymphoma/leukemia (TL/L, *n* = 7), as shown in Table 1. The median age at diagnosis was 66 years (age range of 18–96 years), and in general, the study included more males than females for all tumor entities except for HL, in which the male-to-female distribution was more equivalent. For all patients, pre-therapeutic serum samples were obtained for analysis. For 34 patients diagnosed with either DLBCL (*n* = 23) or FL (*n* = 11), a pre-therapeutic FFPE lymphoma sample was also available for analysis (Appendix A). Moreover, the study included 94 healthy anonymous blood donors who made up the reference group, covering an almost equal distribution of males and females.

### 2.2. sFAP Level and Activity Are Reduced in Aggressive Lymphoid Malignancies

The level of sFAP was measured in the pre-therapeutic serum samples of all patients with lymphoma, yielding a median of 81.0 μg/L (range of 10.8 μg/L–294.2 μg/L). This was comparable to the sFAP levels measured in the samples from the group of healthy individuals, which showed a median of 76.4 μg/L (range of 32.4 μg/L–290.7 μg/L).

As lymphoid malignancies are highly heterogeneous, both in their biology and clinical behavior, we further separated the group of nHLs into subgroups of the aggressive (*n* = 42) or indolent (*n* = 61) entities [19]. Due to missing or incomplete data, seven samples from the “Other B-cell” group were excluded from this distinction. With a median sFAP level of 65.5 μg/L (range of 21.1 μg/L–294.2 μg/L), samples from the aggressive subgroup had significantly decreased levels of sFAP (*p* = 0.002) compared with samples from the indolent subgroup (median = 87.2 μg/L; range of 28.5 μg/L–186.6 μg/L) (Figure 1A). The same observation was made when comparing the samples from the aggressive subgroup with the samples from healthy individuals (*p* = 0.060). Conversely, samples from the indolent subgroup revealed significantly increased sFAP levels (*p* = 0.025) compared with samples from healthy individuals.

Across the individual malignant entities, DLBCL presented with significantly lower sFAP levels than both FL and CLL/SLL (*p* = 0.050 and *p* = 0.006, respectively), which is in accordance with their respective aggressive and indolent characteristics (Appendix A). Moreover, CLL/SLL samples had significantly increased sFAP levels compared with the healthy reference group (*p* = 0.037).

We further investigated sFAP in lymphoid malignancies by measuring its activity across the samples. With a median activity of 832.8 RFU/min (range of 33.0 RFU/min–4609.8 RFU/min), the aggressive subgroup had significantly reduced sFAP activity compared with both the indolent group (median = 1699.7 RFU/min; range of 33.0 RFU/min–5903.3 RFU/min; *p* = 0 < 0.001) and healthy individuals (median = 1444.3 RFU/min; range of 435.6 RFU/min–2712.5 RFU/min; *p* < 0.001) (Figure 1B).

Across the individual subtypes, DLBCL showed significantly lowered sFAP activity compared with the FL, CLL/SLL, and healthy reference groups (*p* = 0.004, *p* = 0.002, and *p* < 0.001, respectively) (Appendix A). Similarly, TL/L showed a general decrease in sFAP activity with lower measurements than the FL, CLL/SLL, and healthy reference groups (*p* = 0.017, *p* = 0.028, and *p* = 0.031, respectively). Interestingly, HL showed a similar pattern of activity as these aggressive entities with significantly reduced sFAP activity compared with FL and CLL/SLL (*p* = 0.029 and *p* = 0.024, respectively) and showed a trend toward reduced activity compared with the healthy refence group (*p* = 0.067) (Appendix A).

Across all included patients, sFAP levels and sFAP activity showed a significant strong positive correlation (*p* < 0.001, ρ = 0.59) (Table 2A). Moreover, both showed significant weaker negative correlations with B symptoms at diagnosis (*p* = 0.046, ρ = −0.18 and *p* = 0.010, ρ = −0.23, respectively), performance score (*p* = 0.013, ρ = −0.22 and *p* < 0.001, ρ = −0.36, respectively), and bulky disease (*p* < 0.001, ρ = −0.31 and *p* = 0.009, ρ = −0.24, respectively). No correlations with other clinicopathological features analyzed were observed. When including only patients with B-cell lymphomas, sFAP levels and activity retained a strong positive correlation (*p* < 0.001, ρ = 0.60), as shown in Table 2B. In addition to the very similar correlations between sFAP levels and activity related to the above-mentioned clinicopathological factors, sFAP activity was also negatively correlated with disease stage and international prognostic index (*p* = 0.015, ρ = −0.28 and *p* = 0.012, ρ = −0.29, respectively).

### 2.3. Ratio of sFAP Activity/sFAP Level Indicates That sFAP Is a Marker of Aggressive Disease

We then considered whether the differences observed between the entities could be attributed to a biological explanation or if the observation of low sFAP activity in the aggressive subgroup was a result of low sFAP levels. To explore this, we calculated the ratio of sFAP activity to sFAP level. Upon this adjustment, the aggressive subgroup continued to show significantly a lower sFAP ratio compared with both the indolent subgroup and the healthy reference group (*p* = 0.012 and *p* = 0.005, respectively) (Figure 1C). No significant differences were observed between the indolent subgroup and the healthy controls after adjustment, which was consistent with the previous results. Moreover, after adjustment, the sFAP activity remained significantly lower in DLBCL, HL, and TL/L patients than in the healthy reference group (*p* = 0.019, *p* = 0.049, and *p* = 0.033, respectively) (Appendix A). A significant difference was also observed indicating a lower ratio in TL/L samples than FL samples (*p* = 0.038).

### 2.4. FAP Expression in DLBCL and FL Tumor Tissues Indicates Opposite Regulation of Serum sFAP Expression

To explore potential correlation between sFAP serum level and FAP tumor tissue expression, we assessed the protein expression levels within lymphoma biopsies from the subgroups of DLBCL (*n* = 23) and FL (*n* = 11) patients from whom a pre-therapeutic lymphoma tissue sample was available (Appendix A). In both DLBCL and FL tissue samples, FAP showed diffuse cytoplasmatic expression of varying degrees, ranging from being almost negatively expressed to strongly positively expressed in the tissue (Figure 2A,B). This heterogeneity in expression levels was especially evident in the DLBCL cases, which may indicate underlying biological differences in FAP regulation or tumor heterogeneity. Interestingly, contrary to the observation in serum, FAP expression was significantly increased in the DLBCL samples compared with the FL samples (*p* < 0.001) (Figure 2B).

We speculated whether FAP expression was affected by the fraction of B cells in the tumor. To investigate this, consecutive tissue sections were stained for the B-cell transcription factor PAX5. As expected, PAX5 staining revealed a general diffuse cytoplasmatic staining of B-cell areas, varying greatly between samples (especially in the DLBCL samples once again), which might mirror different percentages of tumor infiltration in the tissue (Figure 2C,D). Notably, PAX5 expression showed significantly lower levels in the DLBCL samples than the FL samples (*p* = 0.007). Accordingly, we found a significant negative correlation (*p* = 0.022, ρ = −0.38) between the expression levels of FAP and PAX5 in the DLBCL and FL tumor tissues (Table 2C), and FAP expression further showed a negative correlation with bone marrow involvement at the time of diagnosis (*p* = 0.028, ρ = −0.38). Neither FAP nor PAX5 showed significant correlation with any other clinicopathological features analyzed.

Interestingly, when correlating the serum sFAP levels and tumor tissue FAP expression in DLBCL and FL, we observed two interesting patterns (Figure 2E,F). In DLBCL, most patients (78%) presented with higher tumor tissue FAP expression and lower serum sFAP levels. In contrast, in FL, all but one patient (91%) presented with lower tumor tissue FAP and higher serum sFAP levels. Altogether, these differential expression patterns may suggest dysregulated tumor or immune responses and either pathophysiological mechanisms or host response.

### 2.5. Lower sFAP Activity Is Correlated with Inferior Survival Outcome for Aggressive Lymphomas

The median follow-up times were 6.37 years and 6.43 years for the aggressive and indolent subgroups, respectively, with 13 (31%) events of death occurring in the subgroup with aggressive lymphomas within the study period and 12 (20%) occurring in the indolent subgroup. Stratifying patients from the subgroup of aggressive entities according to serum sFAP levels revealed no significant differences in survival outcome (Figure 3A). In contrast, low sFAP activity levels at the time of diagnosis and a low sFAP activity/level ratio were associated with significantly shorter overall survival (OS) for patients with the aggressive entities (both *p* < 0.001) (Figure 3B,C), indicating the significant prognostic value of sFAP activity regardless of sFAP serum levels. Inversely, for the subgroup of indolent entities, sFAP levels, sFAP activity, and sFAP activity/level ratio were not correlated with significant differences in OS (Figure 3D–F).

Moreover, FAP expression in tumor tissues did not show significant associations with survival outcome when analyzing the complete study group nor when exclusively analyzing DLBCL (Appendix A). The same was true for tumor tissue expression of PAX5 (Appendix A). Analyses of outcome were not performed exclusively for FL due to the limited size of this cohort.

## 3. Discussion

This study highlights the potential role of sFAP in lymphoid malignancies, revealing that sFAP may serve as a marker for disease aggressiveness and contribute to tumor biology in hematologic cancers. This study presents the first investigations of sFAP levels and activity in hematologic malignancies and an exploration of intratumoral FAP expression, highlighting important implications for prognostic assessment and treatment strategies.

Particularly striking observations in this study are the differential circulating concentration and enzymatic activity of sFAP across various lymphoma subtypes and the association with the aggressiveness of the disease. We observed that aggressive lymphoid malignancies, including DLBCL and TL/L, exhibited significantly lower sFAP levels and enzymatic activity compared to indolent subtypes like FL and CLL/SLL and to a healthy reference group. Notably, intratumoral FAP expression in pre-therapeutic biopsies from DLBCL and FL patients showed an inverse expression pattern compared to sFAP levels, with higher FAP expression in DLBCL than FL. Altogether, these results suggest that tumoral FAP and sFAP activity in circulation may serve as important biomarkers for disease aggressiveness, potentially reflecting the proteolytic activity within the tumor microenvironment that supports tumor growth and immune evasion.

FAP activity has been reported to be correlated with the concentration of FAP [14]. An intriguing aspect of our study is that sFAP levels alone did not correlate with the survival of patients with the aggressive entities, whereas sFAP activity—along with the ratio of activity to expression—was significantly associated with shorter overall survival in patients with aggressive lymphoma. This might emphasize the importance of functional sFAP activity as a more precise measure of disease biology. While sFAP levels may reflect the general presence or abundance of the protein, in this study, enzymatic activity seemed to directly influence tumor dynamics and patient outcomes. Moreover, Xin et al. [3] suggested that even low FAP enzymatic activity exerts biological effects [3]. In contrast, the lack of correlation between sFAP levels and survival in indolent lymphoma subtypes further reinforces the idea that sFAP activity is more closely linked to aggressive disease phenotypes. The lower sFAP levels observed in aggressive lymphoid malignancies may suggest that the processing and shedding mechanisms of FAP are dysregulated in these malignancies. This may be related to the highly dynamic and variable tumor microenvironment in aggressive lymphomas, where rapid proliferation, immune evasion, and stromal remodeling may alter the balance of proteolytic enzymes like FAP [20]. On the other hand, the relatively higher sFAP levels in indolent lymphomas, particularly CLL/SLL, could reflect a more stable or regulated tumor microenvironment that is important for cancer regulation to support the persistence of a more differentiated or non-aggressive tumor phenotype. In addition to their role in lymphoma subtypes, we found that both sFAP levels and activity were significantly correlated with clinical features of disease severity, such as B symptoms, performance status, and bulky disease. The negative correlation between sFAP levels and performance status suggests that reduced sFAP expression may be indicative of systemic disease involvement and worse functional status, particularly in aggressive lymphoma subtypes.

The origin of sFAP remains controversial, though it is believed that FAP can be shed from the plasma membrane by post translational cleavage, and sFAP is thus composed of the extracellular portion of FAP [2]. While originally described as a cell surface antigen expressed on reactive stromal fibroblasts, subsequent research went on to demonstrate that FAP could also be expressed in non-fibroblast cell types. While FAP is highly expressed in fibroblasts within fibrotic lesions and tumors, its reduced levels in aggressive lymphoid malignancies suggest it may reflect a systemic reaction to the tumor rather than being tumor-derived [2,8,21]. Particularly, our findings support this hypothesis, as distinct patterns of intra-tumoral FAP expression were observed in pre-therapeutic biopsies from DLBCL and FL patients, with DLBCL showing higher FAP expression than FL—an inverse of the sFAP levels. This discrepancy suggests that FAP expression is influenced by both intrinsic and extrinsic factors, such as immune cell in-filtration, stromal components, and hypoxia in the tumor microenvironment. Whether this inverse pattern results from direct tumoral shedding or a systemic host reaction remains unknown. Previous studies of solid tumors have reported a correlation between FAP overexpression (particularly expression in tumor cells) and dismal prognosis [4]. Heterogeneous FAP expression in DLBCL tissues likely contributes to the variable clinical behavior of these tumors. Similar findings were previously reported by other studies, showing higher FAP expression in aggressive lymphomas compared to indolent subtypes [22,23]. Particularly, studies have found that tumoral FAP abundance could be measured in vivo by imaging in the metabolically active lesions [22,23,24,25]. Introducing fibroblast activation protein inhibitor (FAPI) as a PET/CT radiotracer has demonstrated superior uptake and contrast in various solid tumors, attributed to FAPI’s selective targeting of FAP present in the TME. However, the potential utility of FAPI in lymphoma imaging remains relatively unexplored. Preliminary studies have suggested promising though heterogenous results, with selected cases presenting completely absent of FAPI uptake [22,23]. This was especially true in cases of aggressive DLBCLs, and thus, the authors postulate that low FAPI uptake may reflect a distinct subgroup of patients with poorer prognoses characterized by a less reactive stromal microenvironment [24,25]. Specifically, in these cases, low/absent FAPI uptake could identify a group of patients at higher risk of adverse effects, allowing for tailored therapeutic strategies [24,25].

Across lymphoid malignancies, the occurrence, development and progression of the different entities is highly influenced by genetic and environmental factors, including those of the tumor microenvironment [2,15]. Here, we characterized sFAP and its activity across a range of lymphoid malignancies, revealing great heterogeneity. More in-depth analyses are warranted for each separate entity to determine future use of FAP/sFAP as a possible biomarker. Although the use of this intratumoral biomarker in clinical routine has not been protocolized, analyses of FAP in different solid tumors have shown that it has great potential as a prognostic marker, generally being significantly associated with tumor aggressiveness and inferior survival [2,4,8,21,26]. In multiple myeloma, FAP has been shown to activate β-catenin, leading to decreased expression of apoptotic proteins like caspase 3 and protecting tumor cells from chemotherapy-induced cell death [16,17]. This suggests that FAP activation may not only foster tumor growth and survival but also mediate chemoresistance, highlighting its potential as a therapeutic target. Similarly, in acute myeloid leukemia (AML), overexpression of FAP has been linked to poor prognosis, with FAP’s role in activating β-catenin similarly protecting AML cells from apoptosis [18]. This connection underscores FAP as a potential target for therapeutic strategies aimed at overcoming chemoresistance in hematologic malignancies [18].

We observed a negative correlation between FAP expression and the B-cell transcription factor PAX5 in DLBCL, which further supports the notion that FAP may be regulated by factors related to B-cell differentiation and tumor phenotype. This suggests that FAP expression in the tumor microenvironment may not only reflect stromal activation but could also be influenced by the presence of specific immune cell populations or cytokines that modulate fibroblast behavior. The lower levels of PAX5 in DLBCL tissues compared with FL tissues further indicate that the microenvironment in aggressive lymphomas may be less supportive of normal B-cell differentiation, potentially contributing to immune evasion and poorer response to therapy.

Our findings also have important therapeutic implications. Targeting FAP, either directly or by modulating its activity, has emerged as a promising strategy in cancer therapy [2]. Since FAP is overexpressed in the tumor microenvironment and is generally absent from other tissues in healthy adults, some research groups have focused on utilizing FAP protease activity to selectively activate prodrugs at the tumor sites to enhance drug efficacy and reduce toxicity [2,11]. Most candidates are prodrugs that are modified using nanotechnology, although they have not been tested in clinical trials [3]. Depending on the heterogeneous composition of the tumor microenvironment, therapeutic implications may also include cancer-associated fibroblast (CAF)-directed therapy to prevent CAF activation. Further research is needed to determine whether such drugs should be combined with chemotherapy, radiotherapy, targeted therapy, or even immunotherapy. Eliminating CAFs may destabilize the tumor environment and promote metastasis, underscoring the need for further research on FAP-targeted therapies. FAP’s role in suppressing immune responses suggests that it could also boost the effectiveness of immunotherapies. Wang et al. [27] previously suggested FAP as a target for immunotherapy by chimeric antigen receptors (CARs); however, it seems that this strategy has not yet been pursued in clinical therapy. In line with the fast improvements seen in CAR therapy for lymphoma in recent years, further research regarding possible novel biomarkers is warranted. Combining FAP-targeted treatments with immunotherapy warrants further exploration to improve lymphoma outcomes [2,3].

While our study highlights the potential of sFAP as a prognostic biomarker in lymphoid malignancies, several questions remain to be addressed. The interactions between FAP’s enzymatic and non-enzymatic activities, and the molecular pathways through which sFAP influences tumor progression, require further investigation. Recent findings suggest that FAP’s effects on tumor behavior extend beyond its catalytic activity through protein interactions, adding complexity to its role in cancer biology [2]. Future studies should aim to delineate the contributions of these different FAP activities in lymphoma and assess the therapeutic potential of targeting both enzymatic and non-enzymatic functions.

This single-center retrospective study had several limitations, including the small sample sizes of certain subgroups, particularly for FL and TL/L, which limited statistical robustness. Larger studies are needed to validate these findings and assess the clinical utility of sFAP as a biomarker for aggressive disease. While we measured sFAP activity and concentration in serum, further investigations are required to determine whether tissue-specific forms of FAP also play a role in lymphoma pathogenesis. Additionally, the molecular mechanisms driving the observed differences in sFAP levels and activity between lymphoma subtypes remain unclear, warranting more detailed mechanistic studies. Importantly, in the present study, sFAP activity was inferred through its cleavage of FGF21, a hormone with pleiotropic effects on glucose and metabolism as well as cardioprotection activity [14]. FAP exhibits both dipeptidyl peptidase and endopeptidase activities. FGF21 activity is regulated by FAP via proteolytic C-terminal cleavage and thus does not necessarily cover the entire FAP cleave repertoire [14]. A direct comparison of sFAP activity was thus performed under the assumption of undisturbed FGF21 levels, so the results need further validation in future studies.

In conclusion, our study indicates sFAP as a promising biomarker in lymphoid malignancies, with potential utilities in prognostication and treatment stratification. The differential expression and activity of sFAP across aggressive and indolent lymphoma subtypes may reflect underlying biological differences that drive tumor progression and patient outcomes. Future research should focus on further elucidating the mechanistic basis of FAP regulation in lymphoma and evaluating the therapeutic potential of targeting FAP in combination with other treatments.

## 4. Materials and Methods

### 4.1. Patient Samples

For this retrospective study, pre-therapeutic serum samples from 120 patients diagnosed with lymphoma from June 2014 to January 2015 at Aarhus University Hospital were analyzed. The patient cohort covered several different lymphoid malignancies, including DLBCL (*n* = 30), FL (*n* = 13), HL (*n* = 11), CLL/SLL (*n* = 38), TL/L (*n* = 7), and other B-cell nHL not otherwise specified (other B-cell, *n* = 21). Moreover, serum samples from 94 healthy blood donors were used for comparison. For a subset of the DLBCL (*n* = 23) and FL (*n* = 11) patients, pre-therapeutic formalin-fixed, paraffin-embedded (FFPE) lymphoma tissue biopsies were also available for analysis. All biopsies were reviewed by an experienced hematopathologist (SHD) and classified according to the 2017 update on the WHO Classification of Tumours of the Haematopoietic and Lymphoid Tissues [19]. Clinical data were obtained from medical records and have previously been described [28]. The Committees on Health Research Ethics waived the need for written informed consent for the inclusion of participant samples and data in the present study. The study was approved by the Danish National Committee on Health Research Ethics (record no. nr 1-10-72-129-16) and the Danish Data Protection Agency (record nos. 1-16-02-317-16 and 1-16-02-347-13) and was conducted in accordance with the Declaration of Helsinki.

### 4.2. Circulating sFAP Levels and sFAP Activity

Levels of sFAP in serum samples were determined using commercial monoclonal antibodies (DY3715, R&D Systems, Minneapolis, MN, USA) for a modified in-house time-resolved immunofluorometric assay (TRIFMA) as previously described [14,29]. The lowest point of detection (LOD) was 50 ng/L, and the intra- and inter-assay variations (%CV) in the present study were below 11% and 10%, respectively.

The activity of sFAP in serum samples was measured via a modified in-house FRET assay as previously described [14] using the substrate fibroblast growth factor 21 (FGF-21), modified by conjugating both a quencher and a fluorophore to the substrate [14]. The modified substrate was used to determine the ability of sFAP to cleave proteins and thereby its activity [14]. The LOD was 66 relative fluorescent units (RFU)/min. Samples below this limit were assigned half of the LOD value (33.0 RFU/min) The intra- and inter-assay %CV values were below 10% and 11.9%, respectively.

### 4.3. Immunohistochemistry

Immunohistochemical staining for FAP and the B-cell transcription factor PAX5 was performed on 4 µm FFPE tissue sections using the Ventana Benchmark Ultra Automated staining system (Ventana Medical Systems, Tuscon, AZ, USA). Slides were deparaffinized with EZ Prep solution (Ventana, 950–102), followed by blocking of endogenous peroxidase activity using the OptiView DAB IHC Detection Kit (Ventana, 760–700) as previously described [30,31,32,33]. Heat-induced epitope retrieval (HIER) was performed by heating slides at 100 °C for 64 min in ULTRA Cell Conditioning Solution 1 (CC1, Ventana, 950–224). Anti-human FAP (EPR20021, Abcam, Cambridge, UK) primary antibody was diluted to a ratio of 1:50 in REAL^TM^ antibody diluent (DAKO, Glostrup, Denmark, S202230-2) and applied to tissue sections, followed by incubation at 37 °C for 64 min and 8 min of amplification using the Ventana Optiview amplification kit (760–099, Ventana). PAX5 staining was performed by following a previously described protocol [34]. Visualization was performed using the OptiView IHC DAB Detection Kit (Ventana, 760–700) with nuclear counterstaining using hematoxylin. Sections of appendix, tonsil, liver, and pancreas were included on all slides, providing areas of positive and negative controls.

### 4.4. Digital Image Analysis

All stained slides were scanned on the NanoZoomer 2.0HT (Hamamatsu, Shizouka, Japan) at a magnification of 20×. The scanned images were analyzed using the Visiopharm 2020.08 system (Visiopharm A/S, Hoersholm, Denmark) as previously described [33]. In brief, areas of lymphoid tissue suitable for staining quantification were defined by manual outlining of regions of interest (ROIs) on each digitalized section. Analysis protocol packages were designed to quantify the expression levels of each marker as previously described [33]. Staining quantification outputs were area fractions (AFs), calculated as the area of positive staining normalized to the overall area within the ROI.

### 4.5. Statistical Analyses

Differences in sFAP levels, FAP activity, and FAP AFs between patient groups were assessed using an independent Mann–Whitney U test. The correlation of protein expression or activity to clinicopathological features was evaluated using Spearman’s rank test. Outcome analysis was performed using the Kaplan–Meier and log-rank methods with OS as the endpoint. OS was defined as the time from initial FL diagnosis to the date of death by any cause or censoring. For OS analysis, cutoff values for high versus low expression of each biomarker were determined via an ROC analysis, with the optimal cutoff point calculated using Youden’s index. *p*-Values below 0.05 were considered statistically significant. Statistical analyses were performed using R Statistical Software (version 4.1.3).

## Figures and Tables

**Figure 1 ijms-26-11248-f001:**
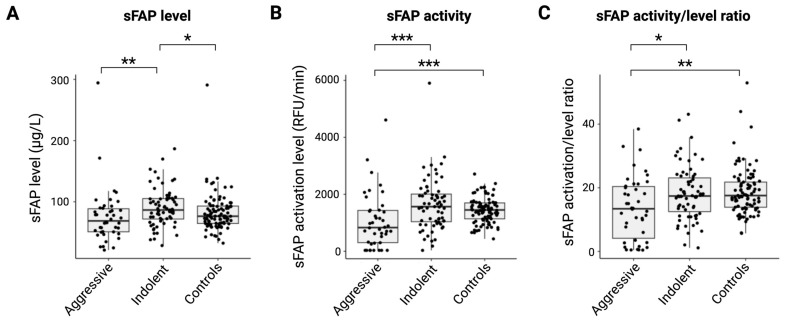
sFAP levels, sFAP activity, and ratio of sFAP activity/level between the aggressive and indolent groups and healthy controls. Presentation of the aggressive group, indolent group, and healthy reference group and their serum (**A**) sFAP levels (μg/L), (**B**) sFAP activity (RFU/min), and (**C**) corrected sFAP activity/sFAP level ratio. * *p* < 0.05; ** *p* < 0.01; *** *p* < 0.001. Abbreviations: RFU, relative fluorescent unit; sFAP, soluble fibroblast activation protein.

**Figure 2 ijms-26-11248-f002:**
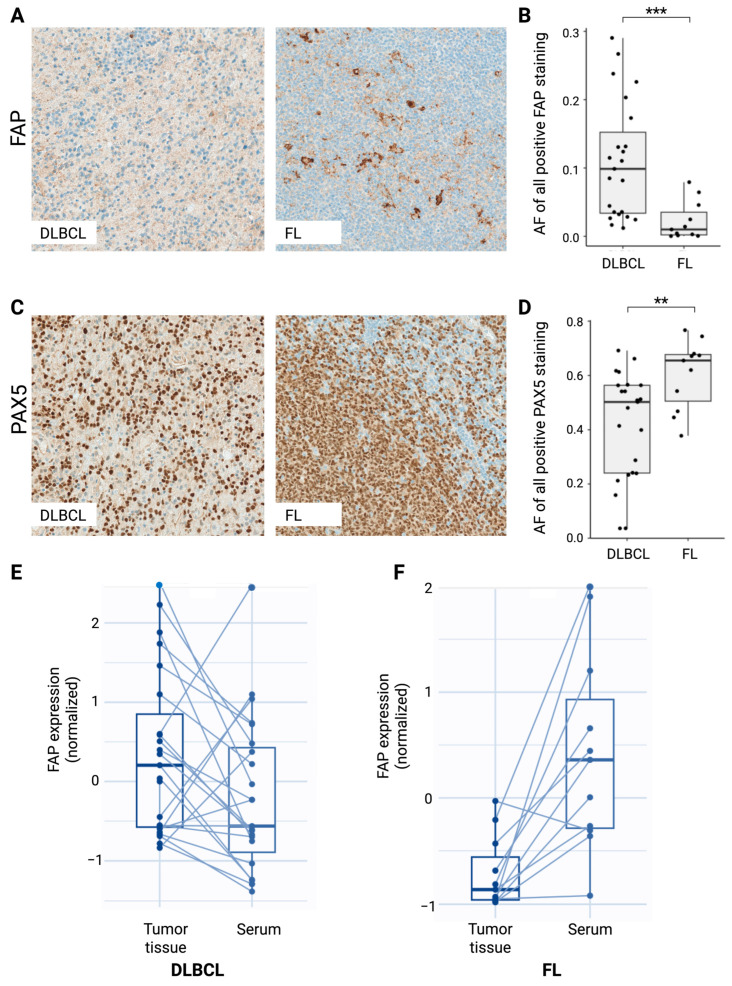
Association between serum sFAP and tumor tissue FAP expression in DLBCL and FL subgroups. (**A**) Representative staining of FAP in DLBCL (left) and FL (right) tumor tissues with magnification of 20×. (**B**) Afs of FAP expression in DLBCL and FL tumor tissues. (**C**) Representative staining of PAX5 in DLBCL (left) and FL (right) tumor tissues with magnification of 20×. (**D**) Afs of PAX5 expression in DLBCL and FL tumor tissues. (**E**,**F**) Correlation between tissue FAP expression and serum sFAP levels in DLBCL (**E**) and FL (**F**). For comparability, the respective expression levels were normalized by Z-scoring. ** *p* < 0.01; *** *p* < 0.001. Abbreviations: AF, area fraction; DLBCL, diffuse large B-cell lymphoma; FL, follicular lymphoma; FAP, fibroblast activation protein; PAX5, paired box 5.

**Figure 3 ijms-26-11248-f003:**
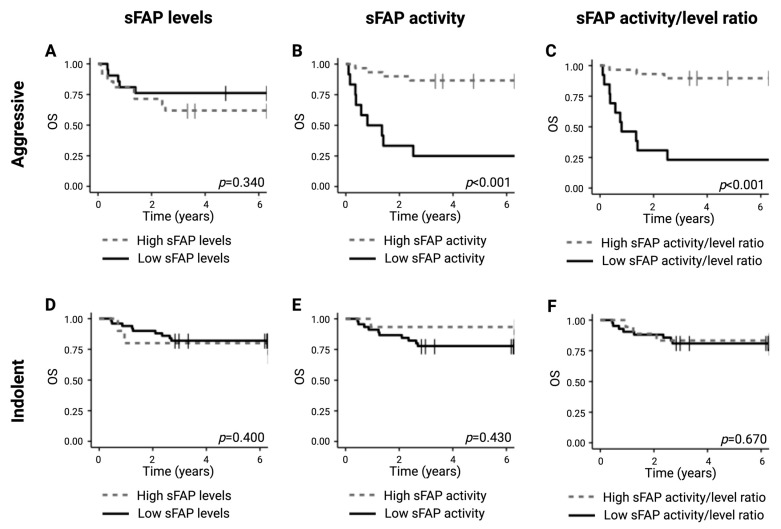
Survival outcome analysis according to sFAP levels, sFAP activity, and ratio in aggressive and indolent lymphoid malignancies. (**A**–**C**) Analysis of OS in the group of aggressive entities according to (**A**) serum sFAP levels (cutoff 65.9 μg/L), (**B**) sFAP activity (cutoff 357.7 RFU/min), and (**C**) sFAP activity/level ratio (cutoff 7.9). (**D**–**F**) Analysis of OS in the group of indolent entities according to (**D**) serum sFAP levels (cutoff 112.6 μg/L), (**E**) sFAP activity (2102.6 RFU/min), and (**F**) sFAP activity/level ratio (cutoff 21.6). Abbreviations: OS, overall survival; sFAP, soluble fibroblast activation protein.

**Table 1 ijms-26-11248-t001:** Patient characteristics.

Characteristics	All Patients (*n* = 120) *n* (%)	DLBCL (*n* = 30, 26%) *n* (%)	FL (*n* = 13, 12%) *n* (%)	HL (*n* = 11, 9%) *n* (%)	CLL/SLL (*n* = 38, 30%) *n* (%)	Other B-Cell (*n* = 21, 17%) *n* (%)	TL/L (*n* = 7, 6%) *n* (%)
Sex							
Male	75 (63)	17 (57)	8 (62)	5 (45)	26 (68)	13 (62)	6 (86)
Female	45 (38)	13 (43	5 (38)	6 (55)	12 (32)	8 (38)	1 (14)
Age at diagnosis, y							
Mean	66	68	65	34	70	63	62
Range	18–96	20–91	33–85	18–70	45–96	49–93	18–78
Comorbidities							
No	27 (23)	6 (20)	4 (31)	7 (64)	6 (16)	2 (10)	2 (29)
Yes	90 (75)	23 (77)	9 (69)	4 (36)	31 (82)	18 (86)	5 (71)
Unknown	3 (3)	1 (3)	0 (0)	0 (0)	1 (3)	1 (5)	0 (0)
Ann Arbor stage							
I–II	27 (23)	8 (27)	5 (62)	6 (55)	-	7 (33)	1 (14)
III–IV	55 (46)	22 (73)	8 (38)	5 (45)	-	14 (67)	6 (86)
Unknown/not relevant	38 (32)	0 (0)	0 (0)	0 (0)	-	0 (0)	0 (0)
B symptoms							
No	72 (60)	14 (47)	11 (85)	3 (27)	30 (79)	12 (57)	2 (29)
Yes	48 (40)	16 (53)	2 (15)	8 (73)	8 (21)	9 (43)	5 (71)
ECOG							
<2	102 (85)	19 (63)	13 (100)	10 (91)	35 (92)	19 (90)	6 (86)
≥2	18 (15)	11 (37)	0 (0)	1 (9)	3 (8)	2 (10)	1 (14)
LDH elevation							
No	80 (67)	18 (60)	12 (92)	6 (55)	29 (76)	12 (57)	3 (43)
Yes	40 (33)	12 (40)	1 (8)	5 (45)	9 (24)	9 (43)	4 (57)
Nodal sites							
<2	89 (74)	7 (23)	8 (62)	7 (64)	34 (89)	3 (14)	2 (29)
≥2	31 (26)	23 (77)	5 (38)	4 (36)	4 (11)	18 (86)	5 (71)
Bulky disease							
No	101 (84)	22 (73)	11 (85)	9 (82)	36 (95)	17 (81)	6 (86)
Yes	13 (11)	6 (20)	2 (15)	2 (18)	0 (0)	3 (14)	0 (0)
Unknown	6 (5)	2 (7)	0 (0)	0 (0)	2 (5)	1 (5)	1 (14)
BM involvement							
No	61 (51)	28 (93)	9 (69)	9 (82)	1 (3)	9 (43)	5 (71)
Yes	57 (48)	2 (7)	3 (23)	1 (9)	37 (97)	12 (57)	2 (29)
Unknown	2 (2)	0 (0)	1 (8)	1 (9)	0 (0)	0 (0)	0 (0)
IPI							
0–I	19 (16)	6 (20)	4 (31)	6 (55)		2 (10)	1 (14)
II–III	41 (34)	11 (37)	7 (54)	4 (36)		14 (67)	4 (57)
IV–V	22 (18)	13 (43)	0 (0)	1 (9)		5 (24)	2 (29)
Unknown/not relevant	38 (32)	0 (0)	2 (15)	0 (0)		0 (0)	0 (0)

BM, bone marrow; CLL/SLL, chronic lymphocytic leukemia/small lymphocytic lymphoma; DLBCL, diffuse large B-cell lymphoma; FL, follicular lymphoma; HL, Hodgkin’s lymphoma; IPI, international prognostic index; LDH, lactate dehydrogenase; TL/L, T-cell lymphoma/leukemia, y, years.

**Table 2 ijms-26-11248-t002:** Correlation between biomarkers and clinicopathological data.

	sFAP Activity	B Symptoms	ECOG	Bulky Disease	Ann Arbor Stage	IPI	BM	PAX5
**(A) All patients**								
sFAP expression	ρ = 0.59 *p* < 0.001	ρ = −0.18 *p* = 0.046	ρ = −0.22 *p* = 0.013	ρ = −0.31 *p* < 0.001	-	-	-	-
sFAP activity	-	ρ = −0.23 *p* = 0.010	ρ = −0.36 *p* < 0.001	ρ = −0.24 *p* = 0.009	-	-	-	-
**(B) B-cell lymphomas**								
sFAP expression	ρ = 0.60 *p* < 0.001	ρ = −0.24 *p* = 0.041	-	ρ = −0.33 *p* = 0.005	-	-	-	-
sFAP activity		ρ = −0.30 *p* = 0.009	ρ = −0.35 *p* = 0.002	ρ = −0.26 *p* = 0.025	ρ = −0.28 *p* = 0.015	ρ = −0.29 *p* = 0.012	-	-
**(C) DLBCL and FL**								
FAP	-	-	-	-	-	-	ρ = −0.38 *p* = 0.028	ρ = −0.38 *p* = 0.022

BM, bone marrow; DLBCL, diffuse large B-cell lymphoma; IPI, international prognostic index; FL, follicular lymphoma; IPI, international prognostic index; PAX5, paired box 5.

## Data Availability

The data presented in this study are available on request from the corresponding author due to Danish/European GDPR legislation.

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
