# Peer review of "Reduced Activity of Soluble Fibroblast Activation Protein (sFAP) Represents a Biomarker of Aggressive Disease in Lymphoid Malignancies"

_ijms, 2025, doi:10.3390/ijms262311248_

Round 1

Reviewer 1 Report

Comments and Suggestions for Authors

This manuscript investigated the role of soluble fibroblast activation protein (sFAP) in lymphoid malignancies and its potential as a biomarker for disease aggressiveness. Through analysis of pre-treatment serum samples from 120 lymphoma patients, the authors found that sFAP levels and activity were significantly reduced in aggressive disease subtypes, and low sFAP activity was associated with shortened overall survival in aggressive entities. Additionally, the study revealed an inverse expression pattern between FAP expression in tumor tissue and serum sFAP. This research provides new insights into the biological significance of FAP in lymphoid malignancies. However, the manuscript still requires improvements in the following areas.

Major Comments

  1. Although the article observed the inverse pattern between serum sFAP and tissue FAP expression, the discussion of underlying mechanisms is relatively superficial. The authors should combine existing literature to provide a more in-depth discussion of the possible biological mechanisms for this inverse expression pattern. Particularly regarding the origin of sFAP, its release mechanisms, and the molecular basis for regulatory differences across different lymphoma subtypes. Recent studies on immune regulatory mechanisms in the tumor microenvironment, the role of protein biomarkers in cancer progression, and genomic approaches in biomarker discovery provide important insights for understanding the complex role of FAP in lymphoid malignancies (doi: 10.1002/mdr2.70007; 10.1016/j.cpan.2024.12.002; 10.1016/j.cpan.2024.12.001; 10.1002/mdr2.70001; PMID: 39225205; 39301900; 38178982).

  1. The authors could discuss in more detail the practical utility of sFAP as a clinical biomarker, including standardization of detection methods, establishment of reference ranges, and comparative advantages over existing clinical indicators. Additionally, they should discuss the cost-effectiveness and feasibility of sFAP testing in clinical practice.

  1. While the article conducted survival analysis, it lacks detailed description of follow-up time and event occurrence rates. The authors should provide key information such as median follow-up time, number of death events, and loss-to-follow-up rates, which are crucial for assessing the reliability of survival analysis results.

  1. The authors could more fully explore in the discussion the similarities and differences between their study results and previous FAP research findings in solid tumors, as well as the significance of these differences for understanding FAP's role in different tumor types. They should also discuss study limitations, particularly the inherent constraints of retrospective design and the generalizability issues of single-center studies.

Minor Comments

  1. The authors should carefully check the full names preceding abbreviations throughout the manuscript to ensure all abbreviations have their complete forms when first introduced.
  2. The article needs professional native English editing to reduce grammatical errors and improve idiomatic expressions.

Reviewer 2 Report

Comments and Suggestions for Authors

                The authors examined the level and activity in the peripheral blood of soluble fibroblast activation protein (sFAP) in patients with different types of lymphoma. A number of interesting differences were observed, warranting further investigation, although that is beyond the scope of this investigation. The study is limited in scope, as is acknowledged by the authors, but is largely well-performed. The examination of serum levels of sFAP is the distinctive contribution of this study, and other aspects from the literature are well-cited. I have two suggestions for improvement of the manuscript:

1) Reference 22 is cited, but simply as similarly having shown higher FAP expression in aggressive lymphomas compared to indolent subtypes. However, there are other aspects of this study that are worthy of mention. FAP was detected by PET imaging with a FAP inhibitor linked to the radionuclide 68Ga, distinct from 18F-labeled FDG simultaneously used for standard metabolic imaging of patients with lymphoma. It would be of interest to readers to know that FAP abundance, and perhaps activity, can be measured in vivo by imaging. Intensities of both tracers were often detected in the same lesions, but there were many instances of lesions that were metabolically active but failed to give evidence of FAP, suggesting heterogeneity that remains to be understood. In comparison to the present manuscript, however, an important difference was that when IHC was performed on samples of lymphoma, FAP was reported to be expressed only in stromal cells, in contrast to metabolic markers (hexokinase and GLUT1) that were observed in neoplastic cells. The authors should comment on this difference.

2) FAP has previously been suggested as a target for immunotherapy with chimeric antigen receptors, from as early as 2014 (PMID: 24778279), but appears not to have been pursued as a clinical therapy. This may be of interest to readers, if the authors choose to mention it.

Round 2

Reviewer 1 Report

Comments and Suggestions for Authors

This manuscript investigated the role of soluble fibroblast activation protein (sFAP) in lymphoid

malignancies and its potential as a biomarker for disease aggressiveness. Through analysis of pre-

treatment serum samples from 120 lymphoma patients, the authors found that sFAP levels and

activity were significantly reduced in aggressive disease subtypes, and low sFAP activity was

associated with shortened overall survival in aggressive entities. Additionally, the study revealed an

inverse expression pattern between FAP expression in tumor tissue and serum sFAP. This research

provides new insights into the biological significance of FAP in lymphoid malignancies. However, the

manuscript still requires improvements in the following areas.

Major Comments

1. Although the article observed the inverse pattern between serum sFAP and tissue FAP

expression, the discussion of underlying mechanisms is relatively superficial. The authors

should combine existing literature to provide a more in-depth discussion of the possible

biological mechanisms for this inverse expression pattern. Particularly regarding the origin of

sFAP, its release mechanisms, and the molecular basis for regulatory differences across

different lymphoma subtypes. Recent studies on immune regulatory mechanisms in the tumor

microenvironment, the role of protein biomarkers in cancer progression, and genomic

approaches in biomarker discovery provide important insights for understanding the complex

role of FAP in lymphoid malignancies (doi: 10.1002/mdr2.70007; 10.1016/j.cpan.2024.12.002;

10.1016/j.cpan.2024.12.001; 10.1002/mdr2.70001; PMID: 39225205; 39301900; 38178982).

2. The article needs professional native English editing to reduce grammatical errors and

improve idiomatic expressions.
